# Fast Bidirectional Probability Estimation in Markov Models

**Siddhartha Banerjee** *
sbanerjee@cornell.edu

**Peter Lofgren**[†]
plofgren@cs.stanford.edu

## Abstract

We develop a new bidirectional algorithm for estimating Markov chain multi-step transition probabilities: given a Markov chain, we want to estimate the probability of hitting a given target state in $\ell$ steps after starting from a given source distribution. Given the target state $t$, we use a (reverse) local power iteration to construct an 'expanded target distribution', which has the same mean as the quantity we want to estimate, but a smaller variance – this can then be sampled efficiently by a Monte Carlo algorithm. Our method extends to any Markov chain on a discrete (finite or countable) state-space, and can be extended to compute functions of multi-step transition probabilities such as PageRank, graph diffusions, hitting/return times, etc. Our main result is that in 'sparse' Markov Chains – wherein the number of transitions between states is comparable to the number of states – the running time of our algorithm for a uniform-random target node is *order-wise smaller* than Monte Carlo and power iteration based algorithms; in particular, our method can estimate a probability $p$ using only $O(1/\sqrt{p})$ running time.

## 1   Introduction

Markov chains are one of the workhorses of stochastic modeling, finding use across a variety of applications – MCMC algorithms for simulation and statistical inference; to compute network centrality metrics for data mining applications; statistical physics; operations management models for reliability, inventory and supply chains, etc. In this paper, we consider a fundamental problem associated with Markov chains, which we refer to as the *multi-step transition probability estimation* (or MSTP-estimation) problem: given a Markov Chain on state space $\mathcal{S}$ with transition matrix $P$, an initial source distribution $\sigma$ over $\mathcal{S}$, a target state $t \in \mathcal{S}$ and a fixed length $\ell$, we are interested in computing the $\ell$-*step transition probability from $\sigma$ to $t$*. Formally, we want to estimate:

$$\mathbf{p}_\sigma^\ell[t] := \langle \sigma P^\ell, \underline{\mathbf{e}}_t \rangle = \sigma P^\ell \underline{\mathbf{e}}_t^T, \tag{1}$$

where $\underline{\mathbf{e}}_t$ is the indicator vector of state $t$. A natural parametrization for the complexity of MSTP-estimation is in terms of the minimum transition probabilities we want to detect: given a desired minimum detection threshold $\delta$, we want algorithms that give estimates which guarantee small relative error for any $(\sigma, t, \ell)$ such that $\mathbf{p}_\sigma^\ell[t] > \delta$.

Parametrizing in terms of the minimum detection threshold $\delta$ can be thought of as benchmarking against a standard Monte Carlo algorithm, which estimates $\mathbf{p}_\sigma^\ell[t]$ by sampling independent $\ell$-step paths starting from states sampled from $\sigma$. An alternate technique for MSTP-estimation is based on linear algebraic iterations, in particular, the (local) power iteration. We discuss these in more detail in Section 1.2. Crucially, however, *both these techniques have a running time of $\Omega(1/\delta)$ for testing if $\mathbf{p}_\sigma^\ell[t] > \delta$* (cf. Section 1.2).

[†]Peter Lofgren is a graduate student in the Computer Science Department at Stanford (http://cs.stanford.edu/people/plofgren/).

## 1.1 Our Results

To the best of our knowledge, our work gives *the first bidirectional algorithm for MSTP-estimation* which works for general discrete state-space Markov chains[1]. The algorithm we develop is very simple, both in terms of implementation and analysis. Moreover, we prove that in many settings, it is order-wise faster than existing techniques.

Our algorithm consists of two distinct *forward* and *reverse* components, which are executed sequentially. In brief, the two components proceed as follows:

- **Reverse-work**: Starting from the target node $t$, we perform a sequence of reverse local power iterations – in particular, we use the REVERSE-PUSH operation defined in Algorithm 1.
- **Forward-work**: We next sample a number of random walks of length $\ell$, starting from $\sigma$ and transitioning according to $P$, and return the sum of residues on the walk as an estimate of $\mathbf{p}_\sigma^\ell[t]$.

This full algorithm, which we refer to as the `Bidirectional-MSTP` estimator, is formalized in Algorithm 2. It works for all countable-state Markov chains, giving the following accuracy result:

**Theorem 1** (For details, refer Section 2.3). *Given any Markov chain $P$, source distribution $\sigma$, terminal state $t$, length $\ell$, threshold $\delta$ and relative error $\epsilon$,* `Bidirectional-MSTP` *(Algorithm 2) returns an unbiased estimate $\widehat{\mathbf{p}}_\sigma^\ell[t]$ for $\mathbf{p}_\sigma^\ell[t]$, which, with high probability, satisfies:*

$$\left| \widehat{\mathbf{p}}_\sigma^\ell[t] - \mathbf{p}_\sigma^\ell[t] \right| < \max\left\{ \epsilon \mathbf{p}_\sigma^\ell[t], \delta \right\}.$$

Since we dynamically adjust the number of REVERSE-PUSH operations to ensure that all residues are small, the proof of the above theorem follows from straightforward concentration bounds.

Since `Bidirectional-MSTP` combines local power iteration and Monte Carlo techniques, a natural question is when the algorithm is faster than both. It is easy to to construct scenarios where the runtime of `Bidirectional-MSTP` is comparable to its two constituent algorithms – for example, if $t$ has more than $1/\delta$ in-neighbors. Surprisingly, however, we show that in *sparse Markov chains* and for *typical target states*, `Bidirectional-MSTP` is order-wise faster:

**Theorem 2** (For details, refer Section 2.3). *Given any Markov chain $P$, source distribution $\sigma$, length $\ell$, threshold $\delta$ and desired accuracy $\epsilon$; then for a uniform random choice of $t \in \mathcal{S}$, the* `Bidirectional-MSTP` *algorithm has a running time of $\widetilde{O}(\ell^{3/2}\sqrt{\overline{d}/\delta})$, where $\overline{d}$ is the average number of neighbors of nodes in $\mathcal{S}$.*

Thus, for typical targets, *we can estimate transition probabilities of order $\delta$ in time only $O(1/\sqrt{\delta})$.* Note that we do not need for every state that the number of neighboring states is small, but rather, that they are small on average – for example, this is true in 'power-law' networks, where some nodes have very high degree, but the average degree is small. The proof of this result is based on a modification of an argument in [2] – refer Section 2.3 for details.

Estimating transition probabilities to a target state is one of the fundamental primitives in Markov chain models – hence, we believe that our algorithm can prove useful in a variety of application domains. In Section 3, we briefly describe how to adapt our method for some of these applications – estimating hitting/return times and stationary probabilities, extensions to non-homogenous Markov chains (in particular, for estimating graph diffusions and heat kernels), connections to local algorithms and expansion testing. In addition, our MSTP-estimator could be useful in several other applications – estimating ruin probabilities in reliability models, buffer overflows in queueing systems, in statistical physics simulations, etc.

## 1.2 Existing Approaches for MSTP-Estimation

There are two main techniques used for MSTP-estimation. The first is a natural Monte Carlo algorithm: we estimate $\mathbf{p}_\sigma^\ell[t]$ by sampling independent $\ell$-step paths, each starting from a random state sampled from $\sigma$. A simple concentration argument shows that for a given value of $\delta$, we need $\widetilde{\Theta}(1/\delta)$ samples to get an accurate estimate of $\mathbf{p}_\sigma^\ell[t]$, irrespective of the choice of $t$, and the structure

of $P$. Note that this algorithm is agnostic of the terminal state $t$; it gives an accurate estimate for any $t$ such that $\mathbf{p}_\sigma^\ell[t] > \delta$.

On the other hand, the problem also admits a natural linear algebraic solution, using the standard power iteration starting with $\sigma$, or the reverse power iteration starting with $\underline{\mathbf{e}}_t$ (which is obtained by re-writing Equation (1) as $\mathbf{p}_\sigma^\ell[t] := \sigma(\underline{\mathbf{e}}_t(P^T)^\ell)^T)$. When the state space is large, performing a direct power iteration is infeasible – however, there are localized versions of the power iteration that are still efficient. Such algorithms have been developed, among other applications, for PageRank estimation [3, 4] and for heat kernel estimation [5]. Although slow in the worst case [2], such local update algorithms are often fast in practice, as unlike Monte Carlo methods they exploit the local structure of the chain. However even in sparse Markov chains and for a large fraction of target states, their running time can be $\Omega(1/\delta)$. For example, consider a random walk on a random $d$-regular graph and let $\delta = o(1/n)$ – then for $\ell \sim \log_d(1/\delta)$, verifying $\mathbf{p}_{\underline{\mathbf{e}}_s}^\ell[t] > \delta$ is equivalent to uncovering the entire $\log_d(1/\delta)$ neighborhood of $s$. Since a large random $d$-regular graph is (whp) an expander, this neighborhood has $\Omega(1/\delta)$ distinct nodes. Finally, note that as with Monte Carlo, power iterations can be adapted to either the source or terminal state, but not both.

For *reversible Markov chains*, one can get a bidirectional algorithms for estimating $\mathbf{p}_{\underline{\mathbf{e}}_s}^\ell[t]$ based on colliding random walks. For example, consider the problem of estimating length-$2\ell$ random walk transition probabilities in a *regular undirected graph* $G(V, E)$ on $n$ vertices [1, 6]. The main idea is that to test if a random walk goes from $s$ to $t$ in $2\ell$ steps with probability $\geq \delta$, we can generate two independent random walks of length $\ell$, starting from $s$ and $t$ respectively, and detect if they *terminate at the same intermediate node*. Suppose $p_w, q_w$ are the probabilities that a length-$\ell$ walk from $s$ and $t$ respectively terminate at node $w$ – then from the reversibility of the chain, we have that $\mathbf{p}_\sigma^{2\ell}[t] = \sum_{w \in V} p_w q_w$; this is also the collision probability. The critical observation is that if we generate $\sqrt{1/\delta}$ walks from $s$ and $t$, then we get $1/\delta$ potential collisions, which is sufficient to detect if $\mathbf{p}_\sigma^{2\ell}[t] > \delta$. This argument forms the basis of the *birthday-paradox*, and similar techniques used in a variety of estimation problems (eg., see [7]). Showing concentration for this estimator is tricky as the samples are not independent; moreover, to control the variance of the samples, the algorithms often need to separately deal with 'heavy' intermediate nodes, where $p_w$ or $q_w$ are much larger than $O(1/n)$. Our proposed approach is much simpler both in terms of algorithm and analysis, and more significantly, it extends beyond reversible chains to any general discrete state-space Markov chain.

The most similar approach to ours is the recent FAST-PPR algorithm of Lofgren et al. [2] for PageRank estimation; our algorithm borrows several ideas and techniques from that work. However, the FAST-PPR algorithm relies heavily on the structure of PageRank – in particular, the fact that the PageRank walk has $Geometric(\alpha)$ length (and hence can be stopped and restarted due to the memoryless property). Our work provides an elegant and powerful generalization of the FAST-PPR algorithm, extending the approach to general Markov chains.

## 2 The Bidirectional MSTP-estimation Algorithm

### 2.1 Algorithm

As described in Section 1.1, given a target state $t$, our bidirectional MSTP algorithm keeps track of a pair of vectors – the estimate vector $\mathbf{q}_t^k \in \mathcal{R}^n$ and the residual vector $\mathbf{r}_t^k \in \mathcal{R}^n$ – for each length $k \in \{0, 1, 2, \ldots, \ell\}$. The vectors are initially all set to $\underline{0}$ (i.e., the all-0 vector), except $r_t^0$ which is initialized as $\underline{\mathbf{e}}_t$. Moreover, they are updated using a *reverse push* operation defined as:

---

**Algorithm 1** REVERSE-PUSH$(v, i)$

---

**Inputs:** Transition matrix $P$, estimate vector $\mathbf{q}_t^i$, residual vectors $\mathbf{r}_t^i, \mathbf{r}_t^{i+1}$

1: **return** New estimate vectors $\{\widetilde{\mathbf{q}}_t^i\}$ and residual-vectors $\{\widetilde{\mathbf{r}}_t^i\}$ computed as:

$$\widetilde{\mathbf{q}}_t^i \leftarrow \mathbf{q}_t^i + \langle \mathbf{r}_t^i, \underline{\mathbf{e}}_v \rangle \underline{\mathbf{e}}_v; \qquad \widetilde{\mathbf{r}}_t^i \leftarrow \mathbf{r}_t^i - \langle \mathbf{r}_t^i, \underline{\mathbf{e}}_v \rangle \underline{\mathbf{e}}_v; \qquad \widetilde{\mathbf{r}}_t^{i+1} \leftarrow \mathbf{r}_t^{i+1} + \langle \mathbf{r}_t^i, \underline{\mathbf{e}}_v \rangle \left( \underline{\mathbf{e}}_v P^T \right)$$

The main observation behind our algorithm is that we can re-write $\mathbf{p}_\sigma^\ell[t]$ in terms of $\{\mathbf{q}_t^k, \mathbf{r}_t^k\}$ as an *expectation over random sample-paths of the Markov chain* as follows (cf. Equation (3)):

$$\mathbf{p}_\sigma^\ell[t] = \langle \sigma, \mathbf{q}_t^\ell \rangle + \sum_{k=0}^\ell \mathbb{E}_{V_k \sim \sigma P^k}\left[\mathbf{r}_t^{\ell-k}(V_k)\right] \qquad (2)$$

In other words, given vectors $\{\mathbf{q}_t^k, \mathbf{r}_t^k\}$, we can get an unbiased estimator for $\mathbf{p}_\sigma^\ell[t]$ by sampling a length-$\ell$ random trajectory $\{V_0, V_1, \ldots, V_\ell\}$ of the Markov chain $P$ starting at a random state $V_0$ sampled from the source distribution $\sigma$, and then adding the residuals along the trajectory as in Equation (2). We formalize this bidirectional MSTP algorithm in Algorithm 2.

---

**Algorithm 2** Bidirectional-MSTP$(P, \sigma, t, \ell_{\max}, \delta)$

---

**Inputs:** Transition matrix $P$, source distribution $\sigma$, target state $t$, maximum steps $\ell_{\max}$, minimum probability threshold $\delta$, relative error bound $\epsilon$, failure probability $p_f$

1: Set accuracy parameter $c$ based on $\epsilon$ and $p_f$ and set reverse threshold $\delta_r$ (cf. Theorems 1 and 2) (in our experiments we use $c = 7$ and $\delta_r = \sqrt{\delta/c}$)

2: Initialize: Estimate vectors $\mathbf{q}_t^k = \underline{0}$, $\forall k \in \{0, 1, 2, \ldots, \ell\}$,
         Residual vectors $\mathbf{r}_t^0 = \underline{\mathbf{e}}_t$ and $\mathbf{r}_t^k = \underline{0}$, $\forall k \in \{1, 2, 3, \ldots, \ell\}$

3: **for** $i \in \{0, 1, \ldots, \ell_{\max}\}$ **do**

4:      **while** $\exists v \in \mathcal{S}$ $\ \ s.t.\ \ \mathbf{r}_t^i[v] > \delta_r$ **do**

5:          Execute REVERSE-PUSH$(v, i)$

6:      **end while**

7: **end for**

8: Set number of sample paths $n_f = c\ell_{\max}\delta_r/\delta$     (See Theorem 1 for details)

9: **for** index $i \in \{1, 2, \ldots, n_f\}$ **do**

10:      Sample starting node $V_i^0 \sim \sigma$

11:      Generate sample path $T_i = \{V_i^0, V_i^1, \ldots, V_i^{\ell_{\max}}\}$ of length $\ell_{\max}$ starting from $V_i^0$

12:      For $\ell \in \{1, 2, \ldots, \ell_{\max}\}$: sample $k \sim Uniform[0, \ell]$ and compute $S_{t,i}^\ell = \ell \mathbf{r}_t^{\ell-k}[V_i^k]$ (We reinterpret the sum over $k$ in Equation 2 as an expectation and sample $k$ rather sum over $k \leq \ell$ for computational speed.)

13: **end for**

14: **return** $\{\widehat{\mathbf{p}}_\sigma^\ell[t]\}_{\ell \in [\ell_{\max}]}$, where $\widehat{\mathbf{p}}_\sigma^\ell[t] = \langle \sigma, \mathbf{q}_t^\ell \rangle + (1/n_f)\sum_{i=1}^{n_f} S_{t,i}^\ell$

---

## 2.2 Some Intuition Behind our Approach

Before formally analyzing the performance of our MSTP-estimation algorithm, we first build some intuition as to why it works. In particular, it is useful to interpret the estimates and residues in probabilistic/combinatorial terms. In Figure 1, we have considered a simple Markov chain on three states – Solid, Hollow and Checkered (henceforth $(S, H, C)$). On the right side, we have illustrated an intermediate stage of reverse work using $S$ as the target, after performing the REVERSE-PUSH operations $(S, 0), (H, 1), (C, 1)$ and $(S, 2)$ in that order. Each push at level $i$ uncovers a collection

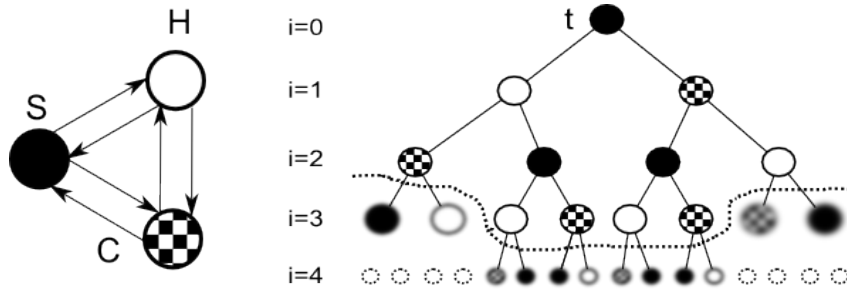

Figure 1: Visualizing a sequence of REVERSE-PUSH operations: Given the Markov chain on the left with $S$ as the target, we perform REVERSE-PUSH operations $(S, 0), (H, 1), (C, 1), (S, 2)$.

of length-$(i + 1)$ paths terminating at $S$ – for example, in the figure, we have uncovered all length 2 and 3 paths, and several length 4 paths. The crucial observation is that *each uncovered path of length $i$ starting from a node $v$ is accounted for in either $\mathbf{q}_v^i$ or $\mathbf{r}_v^i$*. In particular, in Figure 1, all paths starting at solid nodes are stored in the estimates of the corresponding states, while those starting at blurred nodes are stored in the residue. Now we can use this set of pre-discovered paths to boost the estimate returned by Monte Carlo trajectories generated starting from the source distribution. The dotted line in the figure represents the current reverse-work *frontier* – it separates the fully uncovered neighborhood of $(S, 0)$ from the remaining states $(v, i)$.

In a sense, what the REVERSE-PUSH operation does is construct a sequence of importance-sampling weights, which can then be used for Monte Carlo. An important novelty here is that the importance-sampling weights are: $(i)$ adapted to the target state, and $(ii)$ dynamically adjusted to ensure the Monte Carlo estimates have low variance. Viewed in this light, it is easy to see how the algorithm can be modified to applications beyond basic MSTP-estimation: for example, to non-homogenous Markov chains, or for estimating the probability of hitting a target state $t$ for the first time in $\ell$ steps (cf. Section 3). Essentially, we only need an appropriate reverse-push/dynamic programming update for the quantity of interest (with associated invariant, as in Equation (2)).

## 2.3 Performance Analysis

We first formalize the critical invariant introduced in Equation (2):

**Lemma 1.** *Given a terminal state $t$, suppose we initialize $\mathbf{q}_t^0 = \underline{0}, \mathbf{r}_t^0 = \mathbf{e}_t$ and $\mathbf{q}_t^k, \mathbf{r}_t^k = \underline{0} \, \forall \, k \geq 0$. Then for any source distribution $\sigma$ and length $\ell$, after any arbitrary sequence of REVERSE-PUSH$(v, k)$ operations, the vectors $\{\mathbf{q}_t^k, \mathbf{r}_t^k\}$ satisfy the invariant:*

$$\mathbf{p}_\sigma^\ell[t] = \langle \sigma, \mathbf{q}_t^\ell \rangle + \sum_{k=0}^{\ell} \langle \sigma P^k, \mathbf{r}_t^{\ell-k} \rangle \tag{3}$$

The proof follows the outline of a similar result in Andersen et al. [4] for PageRank estimation; due to lack of space, we defer it to our full version [8]. Using this result, we can now characterize the accuracy of the `Bidirectional-MSTP` algorithm:

**Theorem 1.** *We are given any Markov chain $P$, source distribution $\sigma$, terminal state $t$, maximum length $\ell_{\max}$ and also parameters $\delta, p_f$ and $\epsilon$ (i.e., the desired threshold, failure probability and relative error). Suppose we choose any reverse threshold $\delta_r > \delta$, and set the number of sample-paths $n_f = c\delta_r/\delta$, where $c = \max \{6e/\epsilon^2, 1/\ln 2\} \ln (2\ell_{\max}/p_f)$. Then for any length $\ell \leq \ell_{\max}$ with probability at least $1 - p_f$, the estimate returned by `Bidirectional-MSTP` satisfies:*

$$\left| \widehat{\mathbf{p}}_\sigma^\ell[t] - \mathbf{p}_\sigma^\ell[t] \right| < \max \left\{ \epsilon \mathbf{p}_\sigma^\ell[t], \delta \right\}.$$

*Proof.* Given any Markov chain $P$ and terminal state $t$, note first that for a given length $\ell \leq \ell_{\max}$, Equation (2) shows that the estimate $\widehat{\mathbf{p}}_\sigma^\ell[t]$ is an unbiased estimator. Now, for any random-trajectory $T_k$, we have that the score $S_{t,k}^\ell$ obeys: $(i)$ $\mathbb{E}[S_{t,k}^\ell] \leq \mathbf{p}_\sigma^\ell[t]$ and $(ii)$ $S_{t,k}^\ell \in [0, \ell\delta_r]$; the first inequality again follows from Equation (2), while the second follows from the fact that we executed REVERSE-PUSH operations until all residual values were less than $\delta_r$.

Now consider the rescaled random variable $X_k = S_{t,k}^\ell/(\ell\delta_r)$ and $X = \sum_{k \in [n_f]} X_k$; then we have that $X_k \in [0, 1]$, $\mathbb{E}[X] \leq (n_f/\ell\delta_r)\mathbf{p}_\sigma^\ell[t]$ and also $(X - \mathbb{E}[X]) = (n_f/\ell\delta_r)(\widehat{\mathbf{p}}_\sigma^\ell[t] - \mathbf{p}_\sigma^\ell[t])$. Moreover, using standard Chernoff bounds (cf. Theorem 1.1 in [9]), we have that:

$$\mathbb{P}\left[ |X - \mathbb{E}[X]| > \epsilon\mathbb{E}[X] \right] < 2\exp\left( -\frac{\epsilon^2\mathbb{E}[X]}{3} \right) \quad \text{and} \quad \mathbb{P}[X > b] \leq 2^{-b} \text{ for any } b > 2e\mathbb{E}[X]$$

Now we consider two cases:

1. $\mathbb{E}[S_{t,k}^\ell] > \delta/2e$ (i.e., $\mathbb{E}[X] > n_f\delta/2e\ell\delta_r = c/2e$): Here, we can use the first concentration bound to get:

$$\mathbb{P}\left[ \left| \widehat{\mathbf{p}}_\sigma^\ell[t] - \mathbf{p}_\sigma^\ell[t] \right| \geq \epsilon\mathbf{p}_\sigma^\ell[t] \right] = \mathbb{P}\left[ |X - \mathbb{E}[X]| \geq \frac{\epsilon n_f}{\ell\delta_r}\mathbf{p}_\sigma^\ell[t] \right] \leq \mathbb{P}\left[ |X - \mathbb{E}[X]| \geq \epsilon\mathbb{E}[X] \right]$$

$$\leq 2\exp\left( -\frac{\epsilon^2\mathbb{E}[X]}{3} \right) \leq 2\exp\left( -\frac{\epsilon^2 c}{6e} \right),$$

where we use that $n_f = c\ell_{\max}\delta_r/\delta$ (cf. Algorithm 2). Moreover, by the union bound, we have:

$$\mathbb{P}\left[\bigcup_{\ell\leq\ell_{\max}}\left\{\left|\widehat{\mathbf{p}}_\sigma^\ell[t] - \mathbf{p}_\sigma^\ell[t]\right| \geq \epsilon\mathbf{p}_\sigma^\ell[t]\right\}\right] \leq 2\ell_{\max}\exp\left(-\frac{\epsilon^2 c}{32e}\right),$$

Now as long as $c \geq \left(6e/\epsilon^2\right)\ln\left(2\ell_{\max}/p_f\right)$, we get the desired failure probability.

2. $\mathbb{E}[S_{t,k}^\ell] < \delta/2e$ (i.e., $\mathbb{E}[X] < c/2e$): In this case, note first that since $X > 0$, we have that $\mathbf{p}_\sigma^\ell[t] - \widehat{\mathbf{p}}_\sigma^\ell[t] \leq (n_f/\ell\delta_r)\mathbb{E}[X] \leq \delta/2e < \delta$. On the other hand, we also have:

$$\mathbb{P}\left[\widehat{\mathbf{p}}_\sigma^\ell[t] - \mathbf{p}_\sigma^\ell[t] \geq \delta\right] = \mathbb{P}\left[X - \mathbb{E}[X] \geq \frac{n_f\delta}{\ell\delta_r}\right] \leq \mathbb{P}\left[X \geq c\right] \leq 2^{-c},$$

where the last inequality follows from our second concentration bound, which holds since we have $c > 2e\mathbb{E}[X]$. Now as before, we can use the union bound to show that the failure probability is bounded by $p_f$ as long as $c \geq \log_2\left(\ell_{\max}/p_f\right)$.

Combining the two cases, we see that as long as $c \geq \max\left\{6e/\epsilon^2, 1/\ln 2\right\}\ln\left(2\ell_{\max}/p_f\right)$, then we have $\mathbb{P}\left[\bigcup_{\ell\leq\ell_{\max}}\left\{\left|\widehat{\mathbf{p}}_\sigma^\ell[t] - \mathbf{p}_\sigma^\ell[t]\right| \geq \max\{\delta, \epsilon\mathbf{p}_\sigma^\ell[t]\}\right\}\right] \leq p_f$. □

One aspect that is not obvious from the intuition in Section 2.2 or the accuracy analysis is if using a bidirectional method actually improves the running time of MSTP-estimation. This is addressed by the following result, which shows that for typical targets, our algorithm achieves significant speedup:

**Theorem 2.** *Let any Markov chain $P$, source distribution $\sigma$, maximum length $\ell_{\max}$ and parameters $\delta, p_f$ and $\epsilon$ be given. Suppose we set $\delta_r = \sqrt{\frac{\epsilon^2\delta}{\ell_{\max}\log(\ell_{\max}/p_f)}}$. Then for a uniform random choice of $t \in \mathcal{S}$, the* `Bidirectional-MSTP` *algorithm has a running time of $\widetilde{O}\left(\ell_{\max}^{3/2}\sqrt{\overline{d}/\delta}\right)$.*

*Proof.* The runtime of Algorithm 2 consists of two parts:

**Forward-work** (i.e., for generating trajectories): we generate $n_f = c\ell_{\max}\delta_r/\delta$ sample trajectories, each of length $\ell_{\max}$ – hence the running time is $O\left(c\delta\ell_{\max}^2/\delta\right)$ for any Markov chain $P$, source distribution $\sigma$ and target node $t$. Substituting for $c$ from Theorem 1, we get that the forward-work running time $T_f = O\left(\frac{\ell_{\max}^2\delta_r\log(\ell_{\max}/p_f)}{\epsilon^2\delta}\right)$.

**Reverse-work** (i.e., for REVERSE-PUSH operations): Let $T_r$ denote the reverse-work runtime for a *uniform random choice of* $t \in \mathcal{S}$. Then we have:

$$\mathbb{E}[T_r] = \frac{1}{|\mathcal{S}|}\sum_{t\in\mathcal{S}}\sum_{k=0}^{\ell_{\max}}\sum_{v\in\mathcal{S}}(d^{in}(v)+1)\mathbb{1}_{\{\text{REVERSE-PUSH}(v,k)\text{ is executed}\}}$$

Now for a given $t \in \mathcal{S}$ and $k \in \{0, 1, \dots, \ell_{\max}\}$, note that the only states $v \in \mathcal{S}$ on which we execute REVERSE-PUSH$(v, k)$ are those with residual $\mathbf{r}_t^k(v) > \delta_r$ – consequently, for these states, we have that $\mathbf{q}_t^k(v) > \delta_r$, and hence, by Equation (3), we have that $\mathbf{p}_{\underline{\mathbf{e}}_v}^k[t] \geq \delta_r$ (by setting $\sigma = \underline{\mathbf{e}}_v$, i.e., starting from state $v$). Moreover, a REVERSE-PUSH$(v, k)$ operation involves updating the residuals for $d^{in}(v)+1$ states. Note that $\sum_{t\in\mathcal{S}}\mathbf{p}_{\underline{\mathbf{e}}_v}^k[t] = 1$ and hence, via a straightforward counting argument, we have that for any $v \in \mathcal{S}$, $\sum_{t\in\mathcal{S}}\mathbb{1}_{\{\mathbf{p}_{\underline{\mathbf{e}}_v}^k[t]\geq\delta_r\}} \leq 1/\delta_r$. Thus, we have:

$$\mathbb{E}[T_r] \leq \frac{1}{|\mathcal{S}|}\sum_{t\in\mathcal{S}}\sum_{k=0}^{\ell_{\max}}\sum_{v\in\mathcal{S}}(d^{in}(v)+1)\mathbb{1}_{\{\mathbf{p}_{\underline{\mathbf{e}}_v}^k[t]\geq\delta_r\}} = \frac{1}{|\mathcal{S}|}\sum_{v\in\mathcal{S}}\sum_{k=0}^{\ell_{\max}}\sum_{t\in\mathcal{S}}(d^{in}(v)+1)\mathbb{1}_{\{\mathbf{p}_{\underline{\mathbf{e}}_v}^k[t]\geq\delta_r\}}$$

$$\leq \frac{1}{|\mathcal{S}|}\sum_{v\in\mathcal{S}}(\ell_{\max}+1)\cdot(d^{in}(v)+1)\frac{1}{\delta_r} = O\left(\frac{\ell_{\max}}{\delta_r}\cdot\frac{\sum_{v\in\mathcal{S}}d^{in}(v)}{|\mathcal{S}|}\right) = O\left(\frac{\ell_{\max}\overline{d}}{\delta_r}\right)$$

Finally, we choose $\delta_r = \sqrt{\frac{\epsilon^2\delta}{\ell_{\max}\log(\ell_{\max}/p_f)}}$ to balance $T_f$ and $T_r$ and get the result. □

# 3 Applications of MSTP estimation

- **Estimating the Stationary Distribution and Hitting Probabilities:** MSTP-estimation can be used in two ways to estimate stationary probabilities $\pi[t]$. First, if we know the mixing time $\tau_{mix}$ of the chain $P$, we can directly use Algorithm 2 to approximate $\pi[t]$ by setting $\ell_{\max} = \tau_{mix}$ and using any source distribution $\sigma$. Theorem 2 then guarantees that we can estimate a stationary probability of order $\delta$ in time $O(\tau_{mix}^{3/2}\sqrt{\bar{d}/\delta})$. In comparison, Monte Carlo has $O(\tau_{mix}/\delta)$ run-time. We note that in practice, we usually do not know the mixing time – in such a setting, our algorithm can be used to compute an estimate of $\mathbf{p}_\sigma^\ell[t]$ for all values of $\ell \le \ell_{\max}$.

  An alternative is to modify Algorithm 2 to estimate the *truncated hitting time* $\widehat{p}_\sigma^{\ell,hit}[t]$(i.e., the probability of hitting $t$ starting from $\sigma$ for the first time in $\ell$ steps). By setting $\sigma = \mathbf{e}_t$, we get an estimate for the expected *truncated return time* $\mathbb{E}[T_t \mathbb{1}_{\{T_t \le \ell_{\max}\}}] = \sum_{\ell \le \ell_{\max}} \ell \widehat{p}_{\mathbf{e}_t}^{\ell,hit}[t]$. Now, using that fact that $\pi[t] = 1/\mathbb{E}[T_t]$, we can get a lower bound for $\pi[t]$ which converges to $\pi[t]$ as $\ell_{\max} \to \infty$. We note also that the truncated hitting time has been shown to be useful in other applications such as identifying similar documents on a document-word-author graph [10].

  To estimate the truncated hitting time, we modify Algorithm 2 as follows: at each stage $i \in \{1, 2, \ldots, \ell_{\max}\}$ (note: not $i = 0$), instead of REVERSE-PUSH$(t, i)$, we update $\widetilde{\mathbf{q}}_t^i[t] = \mathbf{q}_t^i[t] + \mathbf{r}_t^i[t]$, set $\widetilde{\mathbf{r}}_t^i[t] = 0$ *and do not push back* $\mathbf{r}_t^i[t]$ *to the in-neighbors of $t$ in the $(i+1)^{th}$ stage.* The remaining algorithm remains the same. It is easy to see from the discussion in Section 2.2 that the resulting quantity $\widehat{p}_\sigma^{\ell,hit}[t]$ is an unbiased estimate of $\mathbb{P}[\text{Hitting time of } t = \ell | X_0 \sim \sigma]$ – we omit a formal proof due to lack of space.

- **Exact Stationary Probabilities in Strong Doeblin chains**: A strong Doeblin chain [11] is obtained by mixing a Markov chain $P$ and a distribution $\sigma$ as follows: at each transition, the process proceeds according to $P$ with probability $\alpha$, else samples a state from $\sigma$. Doeblin chains are widely used in ML applications – special cases include the celebrated PageRank metric [12], variants such as HITS and SALSA [13], and other algorithms for applications such as ranking [14] and structured prediction [15]. An important property of these chains is that if we sample a starting node $V_0$ from $\sigma$ and sample a trajectory of length $Geometric(\alpha)$ starting from $V_0$, then *the terminal node is an unbiased sample from the stationary distribution* [16]. There are two ways in which our algorithm can be used for this purpose: one is to replace the REVERSE-PUSH algorithm with a corresponding local update algorithm for the strong Doeblin chain (similar to the one in Andersen et al. [4] for PageRank), and then sample random trajectories of length $Geometric(\alpha)$. A more direct technique is to choose some $\ell_{\max} >> 1/\alpha$, estimate $\{\mathbf{p}_\sigma^\ell[t]\} \ \forall \ell \in [\ell_{\max}]$ and then directly compute the stationary distribution as $\mathbf{p}[t] = \sum_{\ell=1}^{\ell_{\max}} \alpha^{\ell-1}(1-\alpha)\mathbf{p}_\sigma^\ell[t]$.

- **Graph Diffusions:** If we assign a weight $\alpha_i$ to random walks of length $i$ on a (weighted) graph, the resulting scoring functions $f(P, \sigma)[t] := \sum_{i=0}^{\infty} \alpha_i \left(\sigma^T P^i\right)[t]$ are known as a *graph diffusions* [17] and are used in a variety of applications. The case where $\alpha_i = \alpha^{i-1}(1 - \alpha)$ corresponds to PageRank. If instead the length is drawn according to a Poisson distribution (i.e., $\alpha_i = e^{-\alpha}\alpha^i/i!$), then the resulting function is called the *heat-kernel $h(G, \alpha)$* – this too has several applications, including finding communities (clusters) in large networks [5]. Note that for any function $f$ as defined above, the truncated sum $f^{\ell_{\max}} = \sum_{i=0}^{\ell_{\max}} \alpha_i \left(\mathbf{p}_\sigma^T P^i\right)$ obeys $||f - f^{\ell_{\max}}||_\infty \le \sum_{\ell_{\max}+1}^{\infty} \alpha_i$. Thus a guarantee on an estimate for the truncated sum directly translates to a guarantee on the estimate for the diffusion. We can use MSTP-estimation to efficiently estimate these truncated sums. We perform numerical experiments on heat kernel estimation in the next section.

- **Conductance Testing in Graphs:** MSTP-estimation is an essential primitive for conductance testing in large Markov chains [1]. In particular, in regular undirected graphs, Kale et al [6] develop a sublinear bidirectional estimator based on counting collisions between walks in order to identify 'weak' nodes – those which belong to sets with small conductance. Our algorithm can be used to extend this process to any graph, including weighted and directed graphs.

- **Local Algorithms:** There is a lot of interest recently on *local algorithms* – those which perform computations given only a small neighborhood of a source node [18]. In this regard, we note that Bidirectional-MSTP gives a natural local algorithm for MSTP estimation, and thus for the applications mentioned above – given a $k$-hop neighborhood around the source and target, we can perform Bidirectional-MSTP with $\ell_{\max}$ set to $k$. The proof of this follows from the fact that the invariant in Equation (2) holds after any sequence of REVERSE-PUSH operations.

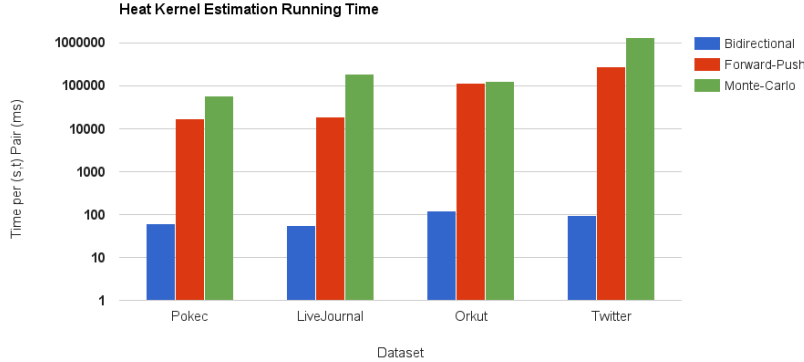

Figure 2: Estimating heat kernels: Bidirectional MSTP-estimation vs. Monte Carlo, Forward Push. To compare runtimes, we choose parameters such that the mean relative error of all algorithms is around 10%. Notice that `Bidirectional-MSTP` is 100 times faster than the other algorithms.

## 4 Experiments

To demonstrate the efficiency of our algorithm on large Markov chains, we use *heat kernel estimation* (cf. Section 3) as an example application. The heat kernel is a non-homogenous Markov chain, defined as the probability of stopping at the target on a random walk from the source, where the walk length is sampled from a $Poisson(\ell)$ Distribution. In real-world graphs, a heat-kernel value between a pair of nodes has been shown to be a good indicator of an underlying community relationship [5] – this suggests that it can serve as a metric for personalized search on social networks. For example, if a social network user $s$ wants to view a list of users attending some event, then sorting these users by heat kernel values will result in the most similar users to $s$ appearing on top. `Bidirectional-MSTP` is ideal for such personalized search applications, as the set of users filtered by a search query is typically much smaller than the set of nodes on the network.

In Figure 2, we compare the runtime of different algorithms for heat kernel computation on four real-world graphs, ranging from millions to billions of edges [3]. For each graph, for random (source, target) pairs, we compute the heat kernel using `Bidirectional-MSTP`, as well as two benchmark algorithms – Monte Carlo, and the Forward Push algorithm (as presented in [5]). All three algorithms have parameters which allow them to trade off speed and accuracy – for a fair comparison, we choose parameters such that the empirical mean relative error each algorithm is 10%. All three algorithms were implemented in Scala – for the forward push algorithm, our implementation follows the code linked from [5].

We set average walk-length $\ell = 5$ (since longer walks will mix into the stationary distribution), and set the maximum length to $\ell + 10\sqrt{\ell} \approx 27$; the probability of a walk being longer than this is $10^{-12}$, which is negligible. For reproducibility, our source code is available on our website (cf. [8]).

Figure 2 shows that across all graphs, `Bidirectional-MSTP` is 100x faster than the two benchmark algorithms. For example, on the Twitter graph, it can estimate a heat kernel score is 0.1 seconds, while the the other algorithms take more than 4 minutes. We note though that Monte Carlo and Forward Push can return scores from the source to all targets, rather than just one target – thus `Bidirectional-MSTP` is most useful when we want the score for a small set of targets.

### Acknowledgments

Research supported by the DARPA GRAPHS program via grant FA9550-12-1-0411, and by NSF grant 1447697. Peter Lofgren was supported by an NPSC fellowship. Thanks to Ashish Goel and other members of the Social Algorithms Lab at Stanford for many helpful discussions.

## Footnotes

*Siddhartha Banerjee is an assistant professor at the School of Operations Research and Information Engineering at Cornell (http://people.orie.cornell.edu/sbanerjee).

[1]Bidirectional estimators have been developed before for *reversible* Markov chains [1]; our method however is not only more general, but conceptually and operationally simpler than these techniques (cf. Section 1.2).

[2] In particular, local power iterations are slow if a state has a very large out-neighborhood (for the forward iteration) or in-neighborhood (for the reverse update).

[3]Pokec [19], Live Journal [20], and Orkut [20] datasets are from the SNAP [21]; Twitter-2010 [22] was downloaded from the Laboratory for Web Algorithmics [23]. Refer to our full version [8] for details.

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
