[Reviews · NeurIPS 2015]

Submitted by Assigned_Reviewer_1

The paper presents a new algorithm for estimating multi-step transition probability (MSTP) for first order time homogeneous Markov chains with finite state space. In the introduction the authors give a clear overview of their results and discusses existing approaches for MSTP estimation. This is followed by a description of their Bidirectional-MSTP algorithm and a theoretical analysis of the algorithm. Finally the authors describe a list of applications of the algorithm and show that their algorithm empirically gives a speed up of at least two orders of magnitude for estimating heat kernels on four standard datasets.

The paper is in general well writing. In particular it is easy to understand the main points in the paper from the introduction. The authors also give an intuitive example explaining their approach (section 2.2).

In terms originality the proposed algorithm is a generalization of FAST-PPR to general Markov chains, as the authors themselves point out. While the time complexities of the algorithms are the same, the proposed Bidirectional-MSTP is substantially simpler than FAST-PPR and is a broad generalization of FAST-PPR.

The significance of the paper is clearly stated through the relative many application of the algorithm within Markov chains, the authors give a least half a dozen relevant examples.

However, I find is somewhat limiting that the authors only evaluate their algorithm on heat kernel estimation. I think it would be appropriate for the authors to give more empirical examples, and in particular also give an example where the algorithm fails.

Minor comments: - On page 2 $n_f$ is used but not explained (at this stage) - In algorithm 1 there is to much space between $q_t^i$ and $+$; and between $r_t^i$ and $-4. - On page 6 I presume that $l_max=P$ should be $l_max = \tau_max$ - On page 7 the following sentence should be revised "... [t]$. for random walks on a (weighted) graph, the such functions called ..." - On page 7: "... the diffusion We perform ..." -> "... the diffusion. We perform ..." - In section 4 you used $t$ for the average walk length, while in the remaining of the manuscript $t$ has been used for the target state.
Summary: The authors present a new algorithm for MSTP estimation that has many applications within Markov models. The simplicity of the algorithm and its many application makes the work significant, but the paper leaves something to be desired in terms of empirical evaluation.

Submitted by Assigned_Reviewer_2

The paper introduces a new algorithm for estimating multi-step transition probabilities (MSTP) for (discrete) Markov chains, and also presents some analysis of the algorithm.

The main theoretical results are (1) a bound on the relative error of the (unbiased) estimate -- this is based on concentration bounds, and (2) a bound on the runtime for sparse problems.

For (2), the interesting result is that the runtime of O(sqrt(1/delta)), where delta is the threshold above which we'd like to estimate MSTPs. The dependence with respect to sparsity is on the average number of neighbors (rather than an upper bound on the number of neighbors). This work appears to build directly on the recent FAST-PPR algorithm (KDD 2014) -- which also achieves runtime O(sqrt(1/delta)) but was designed specifically for PageRank.

Section 3 devotes a page to a bulleted list of potential applications of MSTP estimation.

It feels a bit slapped on, as if to flesh out the paper, but some readers may appreciate this section.

It would be useful here to explicitly distinguish between applications where single MSTP values are of interest and those that call for many or all such values.

This information would complement the experiments Section 4.

I wasn't a fan of the experiments section. The experiments focus exclusively on runtime, but this is not very interesting unless complemented by results for accuracy. It is great that you compare 3 algorithms on 4 real datasets, but you are missing critical experimental results here. I'm also on the fence about whether the comparisons are fair.

As the authors note, the other algorithms "were designed for community detection, and return scores from the source to all targets, rather than just one target."

So, it is indeed interesting that the proposed algorithm here beats the others on a single target (or more generally, a small number of targets).

This suggests that to round out a fair comparison, you might also report how long the bidirectional algorithm would require on all targets.

Alternatively or additionally, you might evaluate a parallel implementation of your algorithm (i.e., the dead-simple embarrassingly parallel thing).

E.g., how many parallel cores would you need to beat the other algorithms on all targets?

- For reasons of interpretability, it would be useful to also report some basic stats for the various datasets in a table -- number of nodes, average degree.

- The bar graphs (Figure 2) on a log scale in units of ms are a bit difficult to interpret. I would consider a table in units of seconds.

Minor comments:

- "order-wise" might be more common

- Some hyphens unnecessary or incorrect:

Monte-Carlo, transition-probabilities, bidirectional-MSTP

- No line break in the middle of "PageRank" (page 2)

- Some of the hyphens in Algorithms seem unnecessary:

estimate-vectors, residual-vectors, sample-path(s)

- Define c as an input to Algorithm 2 (?)

- Avoid using red + green in figures, and it would be nice if figures worked in grayscale

- "we defer it to our full version" -- you mean supplement?

- Use ~ to connect a citation to the previous word

Section 3 looks hastily assembled:

- Could really use some white space (e.g., between bullets)

- For random walks on a (weighted) graph, such functions are called...

- Missing some periods under "Graph Diffusions"

- Footnote 2 (subscript following an integer) is very awkward

Section 4:

- state-of-the-art

Summary: The first half of the paper is a novel algorithm with good theoretical analysis that I would expect to be of interest to members of the NIPS community.

The second half is disappointing and seems hastily slapped on to flesh out the paper.

Author Feedback
Author rebuttal: We thank the reviewers for their time and for the quality of their reviews. We appreciate that many reviewers compliment the simplicity of our algorithm and our analysis. Reviewer criticism focuses on the way we present our applications and experiments, so we'd like to respond to that. We also present a fuller comparison between our work and the algorithm we simplify and generalize.

First Rewiewer_2 raised the question of whether we control for accuracy in our experiments comparing running time. We do control for accuracy, and our submission states that, "For a fair comparison, we choose parameters such that the mean relative error of all three algorithms is around 10%, and for those parameters we measure the mean running time of all three algorithms." Since the mean relative errors are comparable, we omitted them, but we'd be happy to include them in a table in the full version of the paper. Similarly we omitted details on the size of the graphs we use (instead providing a link to the source which has such details), but in response to Reviewer_2's constructive feedback we'd be happy to provide a table describing the graphs we use in the full version of the paper.

Because our algorithm could be used in many diverse current and future applications, it was not easy for us to choose the best applications to describe and to focus on in experiments. To highlight the performance and generalizability of our estimator, we chose heat kernel estimation, a task which is not obviously related to Markov chain multi-step transition probability estimation, but which also has practical relevance to estimating whether a pair of nodes in a network are members of the same community.

Regarding the fairness of experiments, Rewiewer_2 and Reviewer_8 are correct that our algorithm estimates random walk probabilities like the heat kernel between individual (source, target) pairs, rather than from one source to all targets. This is an important difference, and our paper states it explicitly, as Reviewer_2 aptly quotes. Thus our algorithm is relevant in applications where we compute probabilities from a source to one or a few targets. For example consider the problem of re-ranking search results: If a user in a social network like Facebook searched for a common name like "Adam," and Facebook used some other algorithm to find a set of top-k candidates, Facebook may want to boost the rank of candidates which have a high heat-kernel score from the searching user. This is because past work has shown that users in the same community tend to have a higher heat kernel score than other users, and the searching user is presumably more interested in nodes matching the query which are also members of the searcher's community. Similarly, any set of nodes could be personalized in this way, for example when viewing a web page showing an event along with a current set of attending users in some social network, the attending users could be ranked in decreasing heat-kernel score from the viewing user, to highlight members of the same community. In the final version of our paper we'll clarify that our application to heat kernel estimation is most relevant in applications like these involving pairs of node.

Separately, because some reviewers have correctly noted that we simplify and extend FAST-PPR (reference [2] in the paper), we would like to reiterate the improvements of our algorithm relative to FAST-PPR:
- The theoretical bounds obtained in [2] are for a complex variation of FAST-PPR based on 'bouncing random-walks'; FAST-PPR itself has a residual bias and no theoretical accuracy guarantees. Our alternate approach gives a simple estimator which is both unbiased and fast, with a clean analysis.
- The theoretical algorithm in [2] uses structural properties of PageRank including the memoryless property of the walk length distribution, and it is unclear if it extends to the more general MSTP problem.
- FAST-PPR is based on detecting when walks hit a 'frontier set' of nodes - this is not easy in practice, and particularly difficult to parallelize. Our estimate is a simple dot-product - this is very simple to implement and parallelize.

Because our submission presents a fundamentally new way of estimating Markov chain multi-step transition probabilities and a clean analysis of its accuracy and running time, we believe it is a valuable contribution to the NIPS community. We only had space to describe experiments with one application, Heat Kernel estimation between a pair of nodes, but in that application we demonstrate that our algorithm is 100x faster than previous algorithms while controlling for accuracy, showing that our algorithm is efficient in practice. Because of the generality of our result, it has diverse applications, and we expect that more applications of it will arise in the future.